# From Host Defense to Metabolic Signatures: Unveiling the Role of γδ T Cells in Bacterial Infections

**DOI:** 10.3390/biom14020225

**Published:** 2024-02-15

**Authors:** Namya Nanda, Martin P. Alphonse

**Affiliations:** Department of Dermatology, Johns Hopkins University School of Medicine, Baltimore, MD 21231, USA

**Keywords:** γδ T cells, immunometabolism, bacterial infection

## Abstract

The growth of antibiotic-resistant bacterial infections necessitates focusing on host-derived immunotherapies. γδ T cells are an unconventional T cell subset, making up a relatively small portion of healthy circulating lymphocytes but a substantially increased proportion in mucosal and epithelial tissues. γδ T cells are activated and expanded in response to bacterial infection, having the capability to produce proinflammatory cytokines to recruit neutrophils and clear infection. They also play a significant role in dampening immune response to control inflammation and protecting the host against secondary challenge, making them promising targets when developing immunotherapy. Importantly, γδ T cells have differential metabolic states influencing their cytokine profile and subsequent inflammatory capacity. Though these differential metabolic states have not been well studied or reviewed in the context of bacterial infection, they are critical in understanding the mechanistic underpinnings of the host’s innate immune response. Therefore, this review will focus on the context-specific host defense conferred by γδ T cells during infection with *Staphylococcus aureus*, *Streptococcus pneumoniae*, *Listeria monocytogenes*, and *Mycobacterium tuberculosis*.

## 1. Introduction

γδ T cells are an unconventional T cell subset, making up about 1–5% of circulating lymphocytes in most healthy animal species [1]. However, in healthy mucosal or epithelial tissue, these cells can comprise up to 50% of the T cell population [2]. Furthermore, γδ T cell population increases dramatically in response to invading pathogens [3]. Along with Mucosal associated invariant T cells (MAIT), Natural Killer T cells (NKT), and Intraepithelial lymphocytes (IEL), and other unconventional T cell subsets, γδ T cells are non-MHC restricted, can recognize a host of diverse self- and foreign molecules, and have both innate and adaptive immune cell features [4]. 

γδ T cells are classified by their TCRγ and TCRδ chain usage. In humans, this is primarily divided into Vδ1+ and Vδ2+ subsets, which arise during fetal development. Vδ1+ subsets typically associate with the Vγ1+ group, including Vγ2,3,4,5, and 8 [5], and are found in the mucosal epithelium [6,7]. Vδ2+ subsets typically only associate with the Vγ2+ group, including Vγ9 [5], and form the largest γδ T cell population in the peripheral blood [6,7]. The functions of human Vγ9Vδ2+ T cells are well studied, including cytokine production, killing of infected target cells, regulation of monocyte and Dendritic cell (DC) differentiation and maturation, and antigen presentation [8,9,10,11]. 

In mice, however, γδ T cells are classified primarily via their Vγ chain expression and can be resident to specific tissues with non-redundant functions [12]. The first γδ T cell population to arise in the thymus are Vγ5Vδ1 T cells, which migrate to the epidermis and become dendritic epidermal T cells (DETCs), a population not found in humans [13]. DETC progenitors are found between embryonic days 14–16 [14]. In the dermis, however, γδ T cells are primarily Vγ6+, arising at around embryonic day 5, but found obviously on day 3 in mice [12]. These cells are bona fide resident dermal cells and are essential in neonatal skin immunity [12]. Vγ4+ cells increase in number through development [12]. Vγ4+ γδ T cells typically make IFNγ cytokine, while Vγ6+ γδ T cells typically make IL-17 and IL-22 [15]. Thymic signals regulate these cells’ subsequent effector function and critical role during early infection stages [16]. 

γδ T cells may also have memory functions, although it is unclear if these functions are entirely analogous to αβ T cell memory functions [17,18,19,20]. Though not fully elucidated, human Vγ9Vδ2 T cells respond to (E)-4-Hydroxy-3-methyl-but-2-enyl pyrophosphate (HMB-PP), an intermediate in the non-mevalonate (MEP) pathway of isoprenoid synthesis in some pathogenic bacterial species [8,9,21,22]; isopentenyl phosphate (IPP) an intermediate in the mevalonate pathway of isoprenoid synthesis [21]; and superantigens [23]. Other subsets of γδ T cells respond to antigens such as CMV virus and MHC-related molecules [24]. 

γδ T cells are thus important in understanding the host response to bacterial infection. Part of the host immune response includes metabolic reprogramming, coined as immunometabolism, driving subsequent cytokine, chemokine, and interferon responses [25]. This review aims to elucidate the immunometabolic role of γδ T cells during *Staphylococcus aureus*, *Streptococcus pneumoniae*, *Listeria monocytogenes*, and *M. tuberculosis* infections, focusing on their contextual roles in pro- and anti-inflammatory signaling. 

## 2. Immunometabolism of γδ T Cells during Bacterial Infection

Murine γδ T cells can be functionally divided into two subsets via the expression of CD27 and their subsequent metabolic states [26,27] (Figure 1A). CD27+ γδ T cells are generally splenic-resident, ligand-experienced, and secrete IFNγ [26,27]. Alternatively, CD27− γδ T cells are found in the lymph nodes (LNs), are ligand naive, and secrete IL-17 [26,27]. The transcription factor c-Maf is an essential regulator of IL-17 producing γδ T cells by increasing chromatin availability of genes involved in producing Th17 cytokines, *Rorc*, *Il17a*, and *Blk*, and downregulating *TCF1*, which limits IL-17 producing γδ T cell differentiation [28]. These two cell sets have differential metabolic states established during thymic development [29]. IFNγ producing cells are dependent on glycolysis, while IL-17-producing cells rely on oxidative phosphorylation [29], controlled by c-Maf regulation of rate-limiting enzyme isocitrate dehydrogenase 2 (IDH2) [30]. 

While glucose metabolic pathways may mark differential γδ T cell subsets by cytokine production, lipid metabolic changes may impact the severity of inflammation in inflammatory disease (Figure 1B). In a psoriasis model, dietary cholesterol metabolites, such as oxysterol, regulate IL-17+ γδ T cell development and trafficking to the skin, with a higher fat diet in mice being associated with more severe psoriasis [31]. Similarly, IL-17+ γδ T cells had increased lipid metabolism and storage in murine breast adenocarcinoma, colon adenocarcinoma, and melanoma models [29]. 

Amino acid metabolism in T cells has been well studied, particularly in the context of cancer; however, recent studies aim to relate amino acid metabolism in γδ T cells to changes in inflammation severity in psoriasis and polymicrobial sepsis (Figure 1C). For example, glutamine metabolism was upregulated in IL-17+ γδ T cells during psoriasis, promoting acetylation of the *Il17a* promoter and increasing IL-17 production [32,33]. Additionally, CD69 interacted with the heterodimeric LAT1-CD98 amino acid transporter to upregulate tryptophan uptake during psoriasis, leading to mTORC- and AhR-mediated IL-22 production and increased inflammation [33]. Furthermore, glutamine metabolism is also implicated in mouse models of polymicrobial sepsis. Mice treated with glutamine administration had increased γδ T cell populations, decreased inflammatory lesions, and reduced lung damage compared with mice treated with saline [34]. While these metabolic states were established and studied in both an inflammatory and tumor microenvironment, these metabolic changes have not yet been fully elucidated during bacterial infection. To this end, our lab is currently studying immunometabolic states of immune cells, including conventional and unconventional T cell types, including γδ T cell and MAIT cell populations, during *S. aureus* infection. 

## 3. Response of Human Vγ9Vδ2 T Cells to Phosphoantigens

Human Vγ9Vδ2 T cells have remarkably rapid responses to phosphoantigens. IPP, an intermediate in the mevalonate pathway of isoprenoid synthesis, is a stimulator of human Vγ9Vδ2 T cells, though less potent than HMB-PP [24]. Zoledronate (ZOL) and pamidronate (PAM) are aminobisphosphonate drugs and stimulators of human Vγ9Vδ2 T cells [35]. They work by inhibiting farnesyl diphosphate synthase, an enzyme that metabolizes IPP, which leads to an accumulation of IPP, and subsequent activation of Vγ9Vδ2 T cells [35]. The microbial side of γδ T cell immunometabolism largely centers around HMB-PP, one of the most potent stimulators of human Vγ9Vδ2 T cells. HMB-PP is a microbial prenyl phosphate metabolite recognized as a pathogen-associated molecular pattern (PAMP) [35]. To be stimulated by HMB-PP, γδ T cells require butyrophilin membrane proteins [36]. Phosphoantigens like HMB-PP interact with and bind to the B30.2 and juxtamembrane domains within the intracellular domain of BTN3A to modulate phopshoantigen sensing [35]. 

Furthermore, in Vγ9Vδ2 T cells, HMB-PP alone is not sufficient to trigger proliferation; IL-2, IL-17, IL-22, and IL-23 have all been implicated as co-stimulatory cytokines, with IL-23 co-stimulation seen during memory response challenge [36]. This expansion of Vδ2+ cells was found to be a marker for HMB–PP-producing bacterial infection, including *E. coli*, *K. pneumoniae*, *P. aeruginosa*, *C. striatum,* and *L. monocytogenes*. It produced TNFα and IFNγ [8,22,37]. 

HMB-PP stimulated γδ T cells are notably expanded during acute bacterial peritonitis in kidney disease patients [22]. In these patients, HMB-PP stimulated γδ T cells underwent crosstalk with mesothelial cells and peritoneal fibroblasts to induce the proinflammatory response of CCL2, CXCL8, CXCL10, and IL-6 production [22]. These γδ T cells promote short-term neutrophil survival and activation and respond further to neutrophil-phagocytosed bacteria through crosstalk with monocytes [8]. Cell-cell interactions are necessary for such crosstalk, and induce an APC-like phenotype in monocytes, characterized by upregulation of CD40, CD86, and HLA-DR, causing a subsequent inflammatory positive feedback loop [9]. Neutrophils stimulated by Vδ2+ γδ T cells also obtained a unique APC-like phenotype, not found in other circulating neutrophils, again expressing CD40, CD64, CD86, HLA-DR, CD54, and HLA-ABC [38]. 

Interestingly, exposure to TNFα or IFNγ derived from HMB-PP stimulated γδ T cell supernatant induced a morphological change in primary omentum-derived mesothelial cells, causing them to become fibroblastic, coupled with an upregulation of the mesenchymal marker fibronectin [22]. This cellular remodeling may impact the integrity of the peritoneal membrane and could explain increased peritoneal dialysis technique failure rates in patients with bacterial peritonitis [22], making γδ T cell activation by HMB-PP a salient target for therapeutics. Taken in sum, evidence from initial studies marks γδ T cell immunometabolism as a promising field of study in better understanding the microbe and subsequent host response to bacterial invasion. 

## 4. The Site-Specific γδ T Cell Response to *S. aureus* Infection

*S. aureus* is a gram-positive bacterium that is highly pathogenic and is the major cause of skin and soft tissue infection (SSTI), infective endocarditis, bone and joint infection, medical device-related infection, and bacteremia. [39]. Mainly, community-associated methicillin-resistant *S. aureus* (CA-MRSA) infection is a looming threat, claiming over 20,000 lives in the US annually [40]. Furthermore, vaccine efforts against *S. aureus* to date have been unsuccessful [41,42]. As a result, alternative approaches to combat the threat have become essential, and immunotherapies against *S. aureus* infection have emerged in recent years [43,44].

There are various non-pore-forming toxins, pore-forming toxins, and bacterial components from *S. aureus* that are antigenic to the host. γδ T cells have been reportedly involved in early immune responses to *S. aureus* infection. γδ T cells respond to staphylococcal superantigens, including staphylococcal enterotoxin A (SEA) and toxic shock syndrome toxin 1 (TSST-1) [45]. In conventional T cells, staphylococcal superantigens bind outside the MHCII-TCR antigen presenting complex, leading to rapid T cell expansion and inflammation [46]. 

In human adults, Vγ2+ T cell response and proliferation to SEA in specific requires APC presentation, particularly on MHC II [23,47]. SEA recognition requires the N terminal third of the toxin for partial γδ T cell activity and the N terminal two-thirds of the protein for optimal activity, most critically needing amino acid residues 20–27 [23]. Subsequent reactivity to SEA was also dependent on and specific to the Vγ9 region of the TCR, which may explain why Vγ2+ cells are largely extrathymic [47]. 

The γδ T cell cytokine response to superantigens primarily involves IFNγ and IL-17A. IFNγ production by γδ T cells in response to SEA proceeds through an IL-12-dependent pathway and helps induce the expansion of memory-like CD45RO+ Vγ9Vδ2 T cells [41,45]. This finding was echoed in bovine WC1+ γδ T cells [48]. On the other hand, SEA-induced IL-17A production by γδ T cells has a unique role in lung host response. In a lung immunity model, Kumar and colleagues found that γδ T cells were the primary source of proinflammatory cytokine IL-17A after the SEA challenge [49].

Interestingly, however, after the SEA challenge, there was no decrease in neutrophil and monocyte recruitment in TCRδ(−/−) mice, and αβ T cells were found to be responsible for neutrophil and monocyte recruitment to the infection site [49]. Neutrophil recruitment is critical for clearance of *S. aureus*: individuals with impaired neutrophil function or decreased neutrophil count display increased infection susceptibility [50]. Since IL-17A is known to play a critical role in neutrophil recruitment [51], the researchers hypothesized that IL-17A production was being compensated by CD3 + CD8− αβ T cells in TCRδ(−/−) mice [49]. In a follow-up study, IL-17 producing γδ T cells in the lung were termed lung granular γδ T cells due to their increased side scatter measured during flow cytometry analysis [52]. Lung granular γδ T cells were activated by inflammasome-derived IL-1β and IL-2 to produce IL-17 in a JAK/STAT-dependent manner, marking a novel mechanism by which γδ T cells respond to superantigen exposure [52]. Interestingly, this finding contrasts the widely accepted paradigm of IL-2 inhibition of Th17 differentiation [53], though the exact function of γδ T cell-derived IL-17 in this context has not yet been elucidated. 

Unlike SEA stimulation, TSST-1 stimulation of γδ T cells upregulated secretion of proinflammatory cytokines IFNγ, TNFα, and IL-2 and suppressed anti-inflammatory IL-10 production [54]. This response was specific to TSST-1 stimulation and was not demonstrated in other staphylococcal toxins [54]. 

While staphylococcal superantigens were shown to activate the inflammatory response of γδ T cells, *S. aureus* α toxin was contrastingly found to delay IL-17+ γδ T cell recruitment to the infection site, slowing neutrophil recruitment and worsening infection in a mouse dermonecrosis model [55]. These studies indicate a critical role for γδ T cells in activating or suppressing host immune responses against *S. aureus* in response to toxins the bacterium produces. The following section of the review will explore current literature elucidating the host γδ T cell response to *S. aureus* in different infection sites and contexts. 

### 4.1. Cutaneous Infection

Cutaneous γδ T cells are important in early immune defense against *S. aureus* skin infection [56] (Figure 2). In a murine in vivo study of *S. aureus* skin infection, epidermal Vγ5+ γδ T cells induced by IL-1R and IL-23 signaling were protective against worsened infection; γδ T cell deficient mice had more significant lesion sizes, increased bacterial burdens, and lessened neutrophil recruitment and activity compared to both αβ T cell deficient mice and wild type (WT) mice [57]. This protective role is mediated by IL-17 production. γδ T cells comprise most of the IL-17-producing cells in *S. aureus*-infected skin [57,58]. Treatment of γδ T cell-deficient mice with a single dose of IL-17 abrogated the detrimental effects of γδ T cell deficiency [57]. IL-17+ γδ T and Th17 cells also play a compensatory role, promoting neutrophil recruitment in IL-1β deficient mice [59]. 

A study from our research group supported the importance of IL-17 producing γδ T cell populations during *S. aureus* skin infection, finding that these cells were being trafficked from the LNs [60]. T cell receptor (TCR) RNA sequencing revealed clonotypic expansion of Vγ6+ Vδ4+ T cells but not Vγ5+ γδ T cells in the skin and LNs [60]. This expansion was specific to *S. aureus* infection; however, during *P. aeruginosa* infection, both TRGV6 and V5 were expanded [60]. The trafficked Vγ6 + Vδ4+ T cells enhanced the neutrophilic response during secondary *S. aureus* infection in IL-1β deficient mice [61]. These cells produced not only IL-17 but also other proinflammatory cytokines like TNFα, IL-22, and IFNγ [60]. IL-17A and IL-17F produced by LN-trafficked γδ T cells have compensatory roles to one another in the skin, as they do during mucocutaneous *S. aureus* infection [62]. Both cytokines had to be neutralized to note any differences in mouse lesion size or bacterial burden after cutaneous *S. aureus* infection [60]. While IL-17 plays a critical role in γδ T cell signaling in cutaneous infection, it is important to note that this is not the case in all infection contexts. For example, response to early IL-1R signaling, but surprisingly not IL-17 or TNFα signaling, in γδ T cells promotes host survival and monocyte recruitment to the spleen in an *S. aureus*-induced bacteremia model [63].

γδ T cells also have memory-like function during skin inflammation and *S. aureus* infection [61,64]. In an imiquimod model of inflammation, IL-17 producing Vγ4+ γδ T cells leave the LNs and are trafficked to the skin, with previously sensitized mice showing more significant γδ T cell-induced inflammation proliferation and IL-17 production, indicating a memory-like role [64]. In the context of *S. aureus* infection, our colleagues found that while the primary challenge of IL-1β deficient mice led to decreased bacterial clearance, greater lesions, and impaired neutrophil abscess formation, these functions were restored upon secondary infection by LN draining γδ T cells through TLR2/MyD88 signaling to produce IFNγ and TNF [61]. Overall, inflammatory cytokine production in the skin by LN-draining γδ T cells seems to confer protective immunity against primary and secondary *S. aureus* infections. 

Along with inherent host protective responses to *S. aureus* infection, commensal bacteria may protect the host against *S. aureus* infection by driving host antimicrobial peptide production by γδ T cells [65]. One study finds that *S. epidermidis* commensal colonization of the skin induces γδ T cells to upregulate perforin-2 expression, a cytolysin constitutively expressed by γδ T cells to form pores in bacterial membranes [66], along with upregulating other cytotoxicity markers against MRSA infection [67]. As such, *S. epidermidis* co-colonization with *S. aureus* on the skin led to an increased anti-*S. aureus* effect [67]. 

Lactobacilli have also been identified as commensal bacteria that may modulate host immune protection (Figure 2). One study found that in vitro co-colonization of human PBMCs with *S. aureus* and Lactobacillus strains dampened IFNγ secretion in γδ T cells, MAIT cells, and NK cells stimulated by *S. aureus* cell free supernatants [68]. However, recent other studies utilize Lactobacillus recombinant strains as vaccine models against *S. aureus* due to the commensal bacteria’s ability to induce a robust immune response [69,70]. Thus, it is evident that during *S. aureus* infection, lactobacilli have differential effects in vitro versus in vivo and must be studied contextually.

### 4.2. Pneumonia

In the lung, the major subsets of γδ T cells are Vγ1+ and Vγ4+ and accumulate after *S. aureus*-induced pneumonia [71]. While murine γδ T cells are involved in decreasing the bacterial burden and increasing neutrophil infiltration, with corresponding increases in keratinocyte-derived chemokine (KC), MIP2, GM-CSF, IL-6, and TNFα, acute lung damage was decreased in γδ T cell deficient mice, likely due to tamped inflammation from reduced neutrophil recruitment [71]. An early burst of γδ T cell-produced IL-17 was also implicated in increased lung damage post-pneumonia [71], indicating a possibly detrimental role for γδ T cells in this model.

Interestingly, nociceptor sensory neurons in the lung may have an immunosuppressive role on pulmonary γδ T cell function during MRSA infection. Selective ablation of TRPV1+, an ion channel expressed on nociceptors mediating airway allergic pathways, increased absolute Vγ1+ γδ T cell population, increased survival, and increased bacterial clearance [72]. This neuroimmunological finding is important as it marks sensory neurons as targets to protect against *S. aureus* pneumonia. 

Cell death pathways, including the necroptotic pathway, may also suppress the host immune response against *S. aureus*-induced pneumonia by targeting IL-17 signaling by lung γδ T cells [73]. α toxin from *S. aureus* is known to activate the NLRP3 inflammasome in vitro [74]. NLRC4, an NLR family protein involved in inflammasome assembly, is upregulated during *S. aureus*-induced pneumonia in myeloid and non-myeloid cells [73]. Furthermore, α toxin from *S. aureus* induces necroptosis leading to increased IL-18 and IL-1β production and suppressed γδ T cell recruitment, dampening the IL-17 response, leading to decreased neutrophil recruitment and inflammation [73]. However, necroptotic suppression of IL-17 response is site-specific; in a surgical site *S. aureus* infection model, IL-17 producing γδ T cells at the wound site relied by NLRP3/IL-1β signaling for IL-17A production. It is interesting to note that these observed during infection with SH1000 *S. aureus* strain, but not PS80 [37]. 

Therefore, the pneumonia model indicates the necessity of balance in the γδ T cell response; too much IL-17 signaling may lead to excessive inflammation and tissue damage, while too little may hinder bacterial clearance.

### 4.3. Peritonitis

Similar to cutaneous and pulmonary infection, γδ T cells are the primary source of IL-1β-dependent IL-17 in a primary challenge during a recurrent peritonitis murine model of *S. aureus* infection [18]. Some mice in this study were also found to have a biphasic wave of IL-17 production, with one peak at 3 h and the second at 72 h post-infection, with V*γ*4+ γδ T cells at 72 h being primed for later infection and IL-1β independent IL-17 production [18], suggesting a memory function for γδ T cells during acute peritonitis as well. In the kidney, chronic systemic *S. aureus* infection induced the expansion of a population of kidney-resident γδ T cells that constitutively express CD69 and provide protection against *S. aureus* [15]. Thus, in mice, *S. aureus* infection seems to expand both resident and memory γδ T cells. 

In human peritoneal dialysis (PD) patients, on the other hand, it is unclear whether *S. aureus*-induced peritonitis expands γδ T cells or not. In one study, peritoneal Vδ2+ γδ T cells were reduced during acute peritonitis [37]. However, this was not specific to *S. aureus*-induced peritonitis and was used partially as an immune fingerprint to classify gram-positive acute peritonitis [37]. On the other hand, other studies found Vδ2+ γδ T cell recruitment and response to *S. aureus*-induced peritonitis in PD patients, possibly due to superantigen recognition [22,23].

## 5. γδ T Cell Response during *S. pneumoniae* Infection

*S. pneumoniae* infection is the leading cause of community-acquired pneumonia, meningitis, and bacteremia, inflicting a heavy disease burden [75]. Despite vaccination against *S. pneumoniae*, multidrug-resistant bacterial infections run rampant, causing 19,336 hospitalizations in the US annually and millions of dollars in medical and other costs [76]. The immune response to *S. pneumoniae* infection utilizes neutrophil recruitment, subsequent accumulation in the alveolar spaces, and killing through both phagocytosis and, more critically, degranulation [77,78]. Proinflammatory cytokine synthesis and release induced by Vα14+ NKT cells, another unconventional T cell subset, has been shown to promote rapid neutrophil recruitment to the infection site [79]. Alveolar macrophages function to control the resolution of inflammation [80]. γδ T cells have also been critically implicated during innate immune response to *S. pneumoniae* in the lung. 

*S. pneumoniae* infection stimulated human peripheral Vγ9Vδ2 T cells in an in vitro study [81]. In mice, however, Vγ4+ γδ T cells accumulated in the lungs after pulmonary infection [78] due to their lung-homing capability [82]. TCR Vγ4(−/−) mice had increased bacterial burden, increased mortality, decreased neutrophil accumulation, and decreased TNFα and MIP2 production early after *S. pneumoniae* infection, indicating a role for γδ T cells during initial innate immunity [78]. In the lung, Vγ4+ γδ T cells produce IL-17 in response to infection [82]. IL-17 production was stimulated by IL-23 and IL-1β, further recruiting neutrophils [83]. Neutrophils, via NLRP3 inflammasome activation, were in turn the greatest source of IL-1β in the interstitial area of the lung, activated by alveolar macrophage-derived TNFα and pneumolysin, a pore-forming virulence toxin [83]. 

The lung γδ T cell population is regulated to prevent excess inflammation. IL-17-producing γδ T cells are regulated by *S. pneumonia*-activated regulatory T cells with upregulated TNFR2; lack of TNFR2 caused a dysregulated IL-17 response and increased lung damage due to excessive inflammation [84]. The lung’s inflammatory state may also regulate γδ T cell responses [85]. In one study, mice deficient in SOD3, the main enzyme involved in clearing damaging ROS, had increased phagosomal ROS levels [85]. This led to early neutrophil apoptosis in *S. pneumoniae* infection, thereby hampering the subsequent pro-inflammatory γδ T cell response [85]. As such, the above data suggest that γδ T cells have a regulated proinflammatory role during early *S. pneumoniae* clearance in the lung.

γδ T cells have also been found to play an important part in inflammation resolution during *S. pneumoniae* infection. In a murine model, the lung γδ T cell population increased by 30 fold, 7–10 days post-challenge, when no more bacterial burden was detectable [80], indicating that these cells were not involved in bacterial clearance. Instead, lung γδ T cells were cytotoxic towards both naive and challenged host-derived inflammatory alveolar macrophages and pulmonary dendritic cells, thus aiding in dampening inflammation [80], similar to *S. aureus*-induced pneumonia [71]. γδ T cells are also involved in non-bacterial inflammation resolution in the lungs, interacting with M2 macrophages to promote clearance of apoptotic cells during ozone-induced pulmonary inflammation [86]. In summary, early γδ T cell response to *S. pneumoniae* promotes neutrophil recruitment and bacterial clearance, while an end-stage response assists more during the resolution phase of infection.

## 6. γδ T Cell Response during *L. monocytogenes* Infection

*L. monocytogenes* is one of the deadliest food-borne pathogens and can cause listeriosis, which results in a 90% hospitalization rate [87]. Vγ9Vδ2+ T cells have a site-specific proliferative response in the liver and spleen to secreted ligands from the bacterium, indicating a possible role for γδ T cells during immunity [88,89]. HMB-PP is a particularly potent stimulator of human Vγ9Vδ2+ T cells. *L. monocytogenes* undergoes both the mevalonate and non-mevalonate pathways for isoprenoid biosynthesis, producing both IPP and HMB-PP. However, HMB-PP affects a greater response in γδ T cell expansion [90,91]. This is evidenced by the finding that in vitro, human Vγ9Vδ2+ T cells were more bioactive when co-cultured with an *L. monocytogenes* mutant strain with a deficiency in the Lytb (HMB-PP reductase) enzyme leading to HMB-PP overproduction [90]. HMB-PP is also important in developing Vγ9Vδ2 T cells with a memory phenotype, which was demonstrated in a macaque model using an attenuated *L. monocytogenes* immunization [91]. 

γδ T cell activation seems to have a protective role against *L. monocytogenes* infection, though the exact γδ subset involved is unclear. In a murine intraperitoneal infection model, administration of an anti-Vγ1 antibody led to increased bacterial burden in the spleen and liver, indicating a role for Vγ1Vδ6+ T cells during early infection in mice [92]. In contrast, another study found that, despite an overall increase in γδ T cell population after *L. monocytogenes* infection, there was a relative decrease in the frequency of Vγ1+ γδ T cells in the liver, while Vγ6+ T cells increased in frequency [93]. In another inflammation study, Vγ6+ γδ T cells also seemed to have a proportionally increased response compared to other γδ T cell subsets [94]. 

Intravenous injection of *L. monocytogenes* into a lateral tail vein led to γδ T cell activation, which, though non-essential for survival, helped control the inflammatory response in the liver by promoting macrophage influx via MCP-1 production and reducing neutrophil influx [95]. αβ T cells, as opposed to γδ T cells, are critical in survival and protection against necrotic hepatitis [95]. During early infection, IL-17-producing γδ T cells are vital in cytotoxic T cell recruitment. They act on dendritic cells to upregulate MHC I, increasing the production of IL-12, IL-6, and IL-1β, thus leading to the cross-priming and proliferation of CD8+ T cells [59]. In the murine liver, these early IL-17-producing cells are activated by IL-23 from macrophages or dendritic cells, express Vγ4 or Vγ6, and are involved not only in neutrophil recruitment and host protection but also in negatively regulating granuloma formation [59]. Vγ6Vδ1 T cells, in specific, play a protective role during infection and produce IFNγ along with IL-17 [96]. On the other hand, Vγ4+ γδ T cells are protective against CD8+ T cell-mediated liver injury in an IL-10-dependent fashion by controlling TNFα production and dysregulation [97]. IL-10 production in this context occurs in response to signaling from activated splenic macrophages and *L. monocytogenes*-elicited CD8+ T cells. 

γδ T cells have also been shown to have an immunomodulatory response during *L. monocytogenes* infection. Infection in TCRδ deficient mice led to an accumulation of activated macrophages and neutrophils in the infected mice’s peritoneum, leading to overwhelming hepatic necrosis [98], indicating a suppressive role for γδ T cells. Macrophages had decreased apoptosis in TCRδ deficient mice, indicating that γδ T cells are cytotoxic towards macrophages via either the TNFα or the Fas pathway towards the end of *L. monocytogenes* infection [98]. The binding between γδ T cells and macrophages is TCR-mediated [99].

During *L. monocytogenes* infection, γδ T cells also have memory functions. In a mouse oral inoculation model that closely replicates human enteric *L. monocytogenes* infection, mesenteric lymph node-derived Vγ4Vδ1 T cells produced both IL-17A and IFNγ and were identified as a unique CD27^−^CD44^hi^ memory subset that persisted for at least 5 months and launched a robust and rapid proliferative response to secondary challenge [19]. Though it is unclear what causes this memory-like expansion, the secondary memory response was infection route dependent; intravenous infection of mice did not elicit a similar γδ T cell expansion and response [19]. A subsequent study found that Vγ4+ γδ T cells were bona fide resident memory T cells, expressing gut-homing genes like integrin b7, CCR2, CCR5, CXCR3, and CXCR6 [20]. Though these cells had restricted motility, they rapidly formed clusters with neutrophils and monocytes in an IL-17-dependent manner at bacterial replication foci to quickly clear infection after a secondary challenge [20]. A similar γδ T cell population was found in an imiquimod sensitization model of psoriasis, in which they produced elevated levels of IL-17, upregulated IL-1R1 expression, and increasingly responded to IL-1β stimulation [64]. Interestingly, memory Vγ4+ γδ T cells were resistant to cellular senescence [100]. Their adaptive-like response to food-borne *L. monocytogenes* infection increased with age, making these memory T cells an essential target for antibacterial therapies in elderly patients [100]. Thus, γδ T cells have multi-faceted functions, including early protection, inflammatory resolution, and memory functions, during *L. monocytogenes* infection. 

## 7. γδ T Cell Response during *M. tuberculosis* Infection

Tuberculosis is the leading cause of death worldwide from a single infectious agent, primarily due to latent tuberculosis infection [101]. In *M. tuberculosis*-infected lung cells, γδ T cells produce IL-17, dependent upon IL-23 production from infected dendritic cells in mice, and IL-2 co-stimulation and HMB-PP stimulation in primates [36,102,103]. Post expansion, IL-17 producing γδ T cells upregulate expression of ICAM-1 and LFA-1 in macrophages [104,105]. ICAM-1 and LFA-1 are involved in forming granulomas in the lung, thereby helping to sequester and kill the mycobacteria [104,105]. γδ T cell expansion also led to IL-12 production, enhancing the Th1 cytokine response of Ag-specific CD4 and CD8 T cells [36,103]. Macrophages have been implicated in the chemotaxis of γδ T cells to the infection site via secretion of IP-10, inducing changes in chemokine expression pattern [106]; after being trafficked to the pulmonary compartment, Vγ9Vδ2 T cells can differentiate into IFNγ-, perforin-, and granulysin-producing cells, thus reducing bacterial burden and increasing immune resistance against *M. tuberculosis* [36,103]. 

In vitro, HMB-PP-stimulated expansion and effector function are also driven by Th17 cytokines, particularly IL-23 [107]. In one study, IL-2 facilitated IL-23 mediated expansion of HMB-PP stimulated Vγ9Vδ2 T cells from human patients with latent tuberculosis infection [108]. However, tuberculosis infection also selectively impaired IL-23 stimulated expansion during human latent tuberculosis by inhibiting JAK-STAT signaling, leading to loss of effector function and cytokine production [108]. This indicates a mechanism by which *M. tuberculosis* can suppress infection clearance. Specifically, miRNAs hsa-miR-337-3p and hsa-miR-125b-5p were found to be upregulated during tuberculosis infection, inhibiting STAT3 expression and IL-23 stimulated γδ T cell expansion; silencing of these miRNAs restored expansion [108]. *M. tuberculosis* infection may also site-specifically impair Vγ9Vδ2 T cell activation in response to antigen recognition. In human tuberculosis patients, γδ T cells isolated from bronchoalveolar lavage had downmodulation of CD3ε, as compared to γδ T cells from the peripheral blood [109]. Therefore, the *M. tuberculosis* model represents an arms race between the bacteria and the corresponding innate immune response. 

## 8. Future Clinical Role for γδ T Cells

Given the immunoprotective role of γδ T cells, several new therapies have emerged involving stimulation of γδ T cell expansion. γδ T cell-related therapeutics have to date primarily focused on anti-cancer therapeutics. Initially, bisphosphonate-based drugs, such as PAM, ZOL, and IL-2, were found to stimulate Vγ9Vδ2 T cells but were not clinically effective in cancer models [110]. However, due to MHC independence and lessened graft vs. host disease risk, γδ T cells have been identified as a potential target for allogeneic T cell transfer immunotherapy in many cancer states [110,111]. In fact, an ongoing clinical trial is using an expanded γδ T cell infusion intending to increase host defense in leukemia and myelodysplastic patients while reducing the risk of graft vs. host disease seen in conventional bone marrow transplants [110], (Clinical trial ID: NCT03533816). Additionally, induced pluripotent stem cell-derived CAR-modified γδ T cells seem to be effective against tumors in a xenograft mouse model, and there are several commercial entities now developing CAR γδ T cell therapies [110]. A recent murine study found that genetically modified γδ T cells that expressed NY-ESO-1 (a cancer-testis antigen widely expressed in a number of cancers) specific αβTCR are effective in killing NY-ESO-1 expressing tumors [112]. Interestingly, these αβ T cell-transduced γδ T cells seemed to undergo oxidative phosphorylation at a higher rate, with IFNγ production being dependent on ATP production [112]. Mitochondria have been demonstrated to play an important role in sustained CD8 T cell cytotoxicity [113], which may apply to these αβ T cell transduced γδ T cells as well. Though a further examination of the metabolic state of γδ T cells in therapeutic models is required, it could further our understanding of γδ T cell-mediated cytotoxicity, as demonstrated in cancer models.

Apart from cancer models, vaccine development in nonhuman primate tuberculosis models is currently ongoing and focuses on stimulating Vγ9Vδ2 T cells via phosphoantigen delivery. ZOL, in particular, is known to expand γδ T cells, and initial studies focused on adequately promoting γδ T cells to use during immunization. In vitro, ZOL coupled with IL-2 (ZOL/IL-2) stimulation was found to expand IFNγ and TNFα producing γδ T cells [114]. Direct ZOL/IL-2 administration led to an in vivo expansion of Vγ9Vδ2 T cells during multidrug-resistant *M. tuberculosis* infection [36,114]. These cells then traveled to the pulmonary compartment, producing IFNγ, perforin, and TNFα, thereby decreasing bacterial burden and disease state [36,114]. An initial proof of concept study then found that adoptive transfer of ZOL/IL-2-expanded Vγ9Vδ2 T cells in macaques infected with *M. tuberculosis* had reduced bacterial burden in the lung and reduced bacterial dissemination [115]. In another study, a mutant HMB-PP-producing *L. monocytogenes* vector was used to selectively immunize Vγ9Vδ2 T cells, which conferred protection against *M. tuberculosis* in macaques [116]. Vaccine development efforts against *M. tuberculosis* are a promising yet ongoing field of research.

There is also increasing evidence for bisphosphonate-based drugs as an immunotherapy in other bacterial infection states. ZOL activates circulating γδ T cells via monocyte accumulation of IPP and its stereoisomer DMAPP [35,117]. In one study, human septic PBMCs treated with ZOL led to monocyte activation, marked by upregulated HLA-DR, CD40, and CD64, allowing for downstream activation of Vγ9Vδ4 γδ T cells [118]. PAM, a known safe human Vγ9Vδ2 T cell-specific aminobisphosphonate drug [119], may also be a promising therapeutic agent. A chimeric SCID mouse model was used to elucidate the role of memory Vγ9Vδ2 T cells in antibacterial resistance, finding that treatment with PAM rendered PBMCs 100x more efficient in clearing both *S. aureus* and *E. coli* infection [120].

## 9. Conclusions

In summary, this review outlines the critical role of γδ T cells during bacterial infection, focusing on the microenvironment and promising advances in studying immune metabolism. γδ T cells are part in all stages of infection—early defense, inflammatory resolution, and defense during reinfection—making these unconventional T lymphocytes a potential target for therapeutics against hard-to-treat bacterial infection. As such, further research must focus on utilizing γδ T cells in immunotherapies to provide lasting protection against the threat of antibiotic-resistant microbes.

## Figures and Tables

**Figure 1 biomolecules-14-00225-f001:**
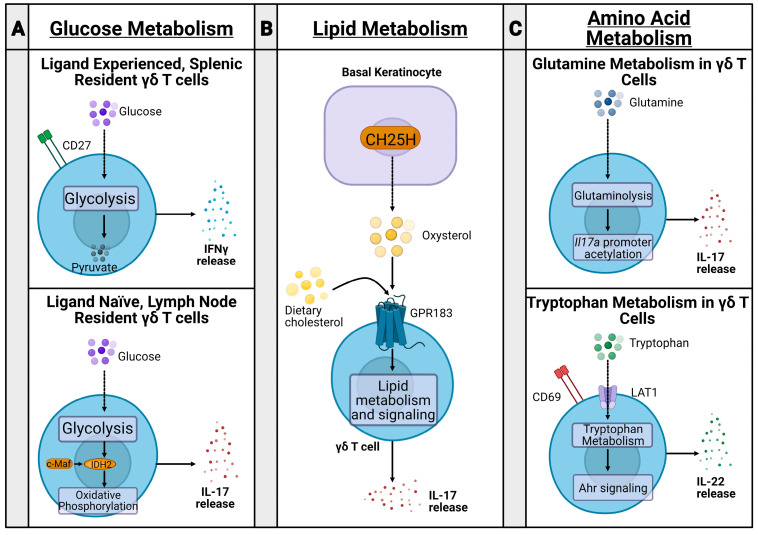
Immune metabolism of γδ T cells. (**A**) CD27+ γδ T cells are ligand-experienced and splenic resident. They undergo glycolysis after thymic development and release IFNγ. CD27− γδ T cells, however, are ligand naive and lymph node resident. They undergo oxidative phosphorylation, governed by c-Maf regulation of rate-limiting enzyme isocitrate dehydrogenase 2 (IDH2) and Th17 cytokine-producing genes *Rorc*, *Il17a*, and *Blk*. (**B**) In inflammatory psoriasis, basal keratinocytes produce oxysterols through the enzyme cholesterol-25-hydroxylase (CH25H). The oxysterols are ligands for the G-protein-coupled receptor GPR183 on γδ T cells. Subsequent lipid metabolism and signaling leads to IL-17 release and inflammation. In a mouse model, dietary cholesterol contributes to this signaling cascade, worsening psoriasis. (**C**) During psoriasis, γδ T cells undergo glutaminolysis, promoting subsequent acetylation of the *Il17a* promoter and releasing IL-17, increasing inflammation. CD69 and LAT1 expression on γδ T cells promote tryptophan uptake and metabolism in psoriasis. The resulting Ahr signaling cascade leads to IL-22 release and increased inflammation.

**Figure 2 biomolecules-14-00225-f002:**
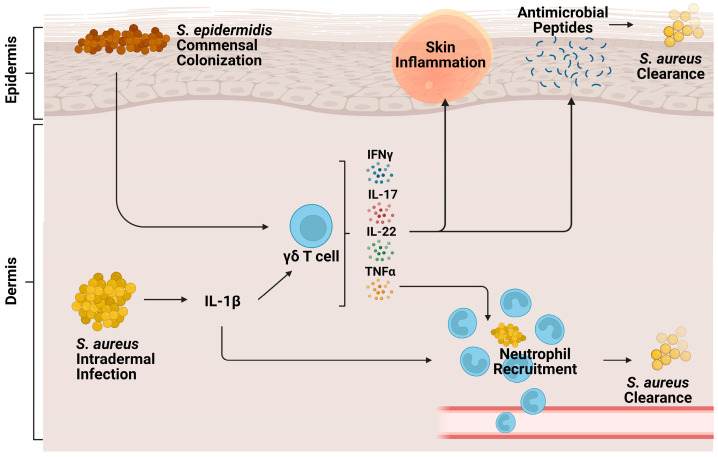
Role of γδ T cells during cutaneous *S. aureus* infection. Intradermal *S. aureus* infection and subsequent IL-1β signaling stimulate cutaneous γδ T cells to produce proinflammatory cytokines IFNy, TNFα, IL-22, and IL-17. These cytokines contribute to neutrophil recruitment, production of antimicrobial peptides, skin inflammation, and eventual bacterial clearance. Commensal *S. epidermidis*, skin colonization, stimulates γδ T cells to produce perforin-2, a pore-forming cytolysin that helps clear *S. aureus* infection.

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
