# Peer review of "From Host Defense to Metabolic Signatures: Unveiling the Role of γδ T Cells in Bacterial Infections"

_biomolecules, 2024, doi:10.3390/biom14020225_

Round 1

Reviewer 1 Report

Comments and Suggestions for Authors

Interesting comprehensive overview of γδT cells with a focus on their (immunometabolic) role in bacterial infections.

Please explain what you mean by "momentum-derived mesothelial cells" (lines 133-134), or rephrase.

Please improve the quality of Fig. 2 (Bacterial clearance).

Author Response

  1. “Please explain what you mean by momentum-derived mesothelial cells (lines 133-134) or rephrase.”

RESPONSE: We thank the reviewer for pointing this out, as this was a spelling error. We made the following change:

Response of Human Vγ9Vδ2 T cells to phosphoantigens

Page 4, lines 146-149: “Interestingly, exposure to TNFα or IFNγ derived from HMB-PP stimulated γδ T cell supernatant induced a morphological change in primary omentum-derived mesothelial cells, causing them to become fibroblastic, coupled with an upregulation of the mesenchymal marker fibronectin [22].”

  1. “Please improve the quality of Figure 2 (Bacterial clearance.)”

RESPONSE: To make the “bacterial clearance” section of Figure 2 more clear, we have revised the label to read “S. aureus clearance” instead. Furthermore, we have included artwork, including icons of cutaneous S. aureus being cleared. Also, we included new directional arrows to indicate the role of IL-1β and TNFα in directly mediating neutrophil recruitment. This neutrophil recruitment is important for the clearance of S. aureus infection in the skin. The updated version of Figure 2 is attached

Reviewer 2 Report

Comments and Suggestions for Authors

Author Response

All revisions are addressed in the attached - pdf 

Round 2

Reviewer 2 Report

Comments and Suggestions for Authors

Major comments have been satisfactorily adressed.

Author Response

Dear Reviewer,

I extend my sincere gratitude for your insightful comments and valuable suggestions. Your review has undoubtedly enhanced the manuscript, enriching its content and quality.

Thank you sincerely,

Martin